# *Cryptosporidium* and *Giardia* in Biogas Wastewater: Management of Manure Livestock and Hygiene Aspects Using Influent, Effluent, Sewage Canal Samples, Vegetable, and Soil Samples

**DOI:** 10.3390/pathogens11020174

**Published:** 2022-01-27

**Authors:** Nguyen Thuy Tram, Pham Duc Phuc, Nguyen Hong Phi, Le Thi Trang, Tang Thi Nga, Hoang Thi Thu Ha, Phung Dac Cam, Tran Quang Canh, Panagiotis Karanis

**Affiliations:** 1Department of Bacteriology, National Institute of Hygiene and Epidemiology (NIHE), 1 Yersin, Hanoi 122000, Vietnam; ntt3@nihe.org.vn (N.T.T.); letrang1980@gmail.com (L.T.T.); tangngah@gmail.com (T.T.N.); htth@nihe.org.vn (H.T.T.H.); phungdaccam1@gmail.com (P.D.C.); 2Center for Public Health and Ecosystem Research (CENPHER), Hanoi University of Public Health, 1A Duc Thang, Hanoi 122000, Vietnam; pdp@huph.edu.vn (P.D.P.); nhp@huph.edu.vn (N.H.P.); 3Department of Medical Laboratory Science, Hai Duong Medical Technical University, 1 Vu Huu, Thanh Binh, Hai Duong 17000, Vietnam; canhhdt@gmail.com; 4Medical Faculty, University of Cologne, University Hospital, 50931 Cologne, Germany; 5Department of Basic and Clinical Sciences, Medical School, Anatomy Institute, University of Nicosia, Nicosia 2408, Cyprus

**Keywords:** biogas wastewater, *Giardia*, *Cryptosporidium*, detection, Vietnam

## Abstract

*Cryptosporidium* and *Giardia* are two water- and foodborne protozoan parasites that can cause diarrheal diseases. Poor microbial quality, sanitation conditions, and hygiene practices at exposure to biogas wastewater are important risk factors for human and animal infection. This study highlights the presence and level of both parasites in the environment in relation to biogas waste reuse in Vietnam. A total of 239 samples were collected from different types of samples in the studied districts in Bac Giang province in 2020 via direct immunofluorescent detection to study the occurrence of *Cryptosporidium* spp. and *Giardia* spp. (oo)cysts. Among the samples, *Cryptosporidium* was found in 19 (7.9%) with concentration from 1.10^4^ to 3.10^5^ oocysts/100 mL, while *Giardia* in 40 (16.7%) with concentration from 1.10^4^ to 2.10^6^ cysts/100 mL, respectively. In detail, the results show that the percentages of positive detection of *Cryptosporidium* spp. and *Giardia* spp. in influent, effluent, sewage canal, and vegetables were 13.1% (11/84), 6.0% (5/83), 15.4% (2/13) and 5.9% (1/17) and 26.2% (22/84), 7.2% (6/83), 7.7% (1/13) and 5.9% (1/17), respectively. The results show a trend of decreasing *Cryptosporidium* and *Giardia* densities, without statistical significance. Although these parasites decreased after biogas treatment, the remaining loads observed in biogas effluent can reach the watercourses and soil receiving it. Further investigations are needed to contribute to a general understanding of the risk of protozoan parasites, as well as strategies to control and reduce the contamination of environmental water sources and plants and reduce the burden of the pathogens in biogas wastewater in Vietnam.

## 1. Introduction

Agricultural production plays a leading role in Vietnam. Livestock and poultry production have been gradually changing in scale from household to farm. Along with the development of the livestock sector, biogas technology has helped livestock farmers in treating animal waste and providing clean energy to the community. Building and using biogas plants in livestock production are one of the effective solutions for managing large volumes of animal manure. Biogas technology minimizes the negative impact of pollution on the environment, reduces disease risks for humans, animals, and diminishes the risk of livestock waste in agricultural areas. At the same time, it contributes to the sustainable development of household husbandry and farming, the generation of clean energy sources, the improvement of livelihoods, and the enhancement of the quality of life for rural people, all of which are in line with the current development trend of the agricultural industry [1]. With these many proven benefits, biogas technology has become widespread throughout Asia and Vietnam.

The discharge of raw wastewater into the environment due to poor management and insufficient treatment of livestock manure is undoubtedly a contamination source of bacterial, fungal, viral, and protozoa pathogens. Among these parasites, *Cryptosporidium* spp. and *Giardia* spp., are transmitted through oocysts and cysts which are excreted in high numbers in the feces of infected hosts [2,3]. Consequently, raw water bodies, soils, and vegetables are subject to parasitic contamination. Moreover, (oo)cysts are resistant to environmental stresses, which makes their disposal using conventional methods and disinfection processes difficult [3]. Taking into consideration the low infective dose of these parasites, it has been suggested that exposure of wastewater could infer a health risk of protozoan infections [4]. As a result, these parasites are responsible for several outbreaks worldwide [5,6,7,8,9], causing them to be emerging and re-emerging targets of current research on food and waterborne parasites [10].

Several studies have identified high levels of *Cryptosporidium* and *Giardia* oo(cysts) in both treated and untreated wastewater from Wastewater Treatment Plants (WWTPs) [11,12,13]. An extended preliminary survey of biological treatment plants indicated that *Cryptosporidium* oocysts are found both in the input and the discharge of the biological treatment plants in Greece [14]. Findings from a study in Canada indicated that anaerobically digested biosolids can contain up to 10^1^ oocysts/g of *Cryptosporidium* and 10^2^ cysts/g of *Giardia* [15].

The occurrence of the (oo)cysts of *Cryptosporidium* and *Giardia* in agricultural water and other aquatic environments is a globally acknowledged public health problem. An European surveillance of communicable disease consistently describes the occurrence of significant numbers of verified cases of cryptosporidiosis and giardiasis that are distributed across virtually all of the countries in and adjacent to Europe [16]. This highlight features of their presence and distribution that present a clear risk of waterborne transmission. In addition, this should dictate features of monitoring essentials to minimize risk through the effective management of watersheds and water treatment systems.

In Vietnam, household-scale anaerobic digesters from livestock manure are currently promoted for bioenergy production as they reduce odor and provide an effluent with a high fertilizer value that can be used to fertilize field, garden crops, and fishponds [17]. However, biogas effluents can pollute aquifers, crops, and surface water due to inadequate manure management. As a result, risks of environmental pollution are increased, and high incidences of giardiasis and cryptosporidiosis may be attributed to livestock waste. Two investigations [18,19] reported a high prevalence of protozoan parasites (*Cryptosporidium*, *Cyclospora*, *Giardia*) in environmental samples collected from farms and markets in Hanoi and Hanam, Vietnam. Detection of zoonotic species had confirmed the potential high risk of protozoan infection to those who have been exposed to wastewater, contaminated soil, water, and food. In a previous study, Le-Thi et al. in 2017 [20] estimated that the annual diarrhea risk caused by exposure to biogas effluent through irrigation activities ranges from 17.4 to 21.1% (*E. coli* O157:H7), 1.0 to 2.3% (*G. lamblia*), and 0.2 to 0.5% (*C. parvum*), while those caused through unblocking drains connected to biogas effluent tanks were 22% (*E. coli* O157:H7), 0.7% (*G. lamblia*), and 0.5% (*C. parvum*). Although studies in Vietnam have not found evidence of *Cryptosporidium* and *Giardia* clinical diseases among children with diarrhea [21,22,23], it is most likely that these parasites are widely distributed in these populations. A study on HIV positive humans found that the *C. parvum* human genotype infects three people in Vietnam [24]. Healthy people (N = 2522) in north-western Vietnam had a surprisingly high prevalence of *Giardia* (4.1%) [25]. Recently, a total of 2715 samples (2120 human diarrheal samples, 471 human non-diarrheal samples, and 124 animal stool samples) were collected from a community survey in an agricultural area in Vietnam. It was found that 15 samples (10 diarrheal samples, 2 non-diarrheal samples, and 3 animal stool samples) tested positive with *Cryptosporidium* and (23 animal stool samples, 8 human non-diarrheal samples, and 36 human diarrheal samples) with *Giardia* by PCR [26,27].

Considering the occurrence and contamination level of *Cryptosporidium* and *Giardia* (oo)cysts as well as hygiene aspects in the management of biogas wastewater, this study aimed to determine the occurrence and evaluate the levels of *Giardia* and *Cryptosporidium* (oo)cysts in pre- and post-biogas waste treatment that are used for agriculture activities in Vietnam. Obtaining data on the quality of biogas wastewater will provide a better understanding on the current sanitation of biogas plants, which are valuable information for further research on the intervention and assessment of human health risk of communities with biogas plants installation.

## 2. Materials and Methods

### 2.1. Study Area

Bac Giang province is located about 80 km east of Hanoi (population 1,841,000) (Figure 1). The Bac Giang livestock sector is developing to contribute to the provincial value of the agricultural sector in Vietnam. Pig farming is mainly located in Tan Yen, Lang Giang, Viet Yen, and Luc Ngan districts. Poultry is mainly concentrated in districts of Yen The, Tan Yen, Luc Ngan, Luc Nam, Hiep Hoa, and Lang Giang. Ducks are mainly raised in Lang Giang, Hiep Hoa, and Yen The districts. The total number of households raising chicken in Bac Giang province is 237,387, accounting for 83.7% of agricultural production households and 63.47% of total rural households. Due to the strong development of livestock, changes have occurred in livestock production due to the establishment of large-scale farms. Along with the development of the livestock sector, biogas technology has helped livestock farmers in treating animal waste and providing clean energy to the community. Using biogas technology is an effective solution to the problems of supplying energy and reducing environmental pollution in the countryside. However, in Thai Binh and Bac Giang provinces, 43% of pig manure is used as biogas substrate, with 19% of the untreated and composted manure discharged into aquatic recipients and 17% of the biogas effluent discharged into rivers or canals [17]. In addition, many farmers discharge biogas effluent into the environment, e.g., the household garden, canals, lakes, and occasionally the public sewer system, as many farmers are not aware of the nutrient content and fertilizer value of biogas effluent. Hiep Hoa, Lang Giang, Yen The, and Viet Yen are the districts with the largest number of biogas plants in Bac Giang, accounting for nearly 70% of biogas plants built in the province. Most of the biogas plants have been built as KT1 and KT2 types (Figure 2) [1]. Therefore, the study was conducted in Lang Giang, Viet Yen, and Hiep Hoa districts. These districts are peri-urban areas of Bac Giang city. Biogas waste was used mainly for fruit tree, corn, and vegetable cultivation. The districts had more than 97% of farmers. There was a mix between farming families exposed to biogas wastewater and those who were not exposed, depending on the installation of a biogas plant at the households.

### 2.2. Study Population

Three hundred and ninety-four households (135 households in Lang Giang, 172 households in Viet Yen, and 87 households in Hiep Hoa) were randomly selected from an official list of households from each district. Households that had not installed a biogas plant and were not involved in agriculture at all were excluded. All of the females and males with 15 years of age and above living in the selected households were included in the questionnaire interviews.

### 2.3. Household Questionnaire

A cross-sectional study was conducted from March through August 2020. The selected households were surveyed using a structural questionnaire administered to the head of the household or their partner. The extent of contact with biogas wastewater was assessed for all of the individuals (≥15 years of age and living in the study area at the time of the study). The extent of contact included the status of any contact with biogas wastewater and the frequency of contact (hours per contact, days of contact per month etc.). This information was obtained from the person him/herself or from the main respondent of the household in the case that the person was absent at the time of the interview. The following information was also collected for each individual: Age, sex, occupation, educational level, types of agriculture production involved, and the use of protective measures during agricultural work. At the household level, information was obtained on household source and sanitation (hygienic status and biogas waste handling), livestock manure use practices in agriculture/horticulture, and personal hygiene behavior including hand washing practices [18].

### 2.4. Sample Collection and Protozoan Parasitological Analyses

A total of 239 samples including 84 influent samples, 83 effluent samples, 13 sewage canal samples, 21 pond/lake water samples, 21 soil samples, and 17 vegetables were collected from six locations. Samples were collected during the household survey between March through August 2020 (Figure 3).

#### 2.4.1. Collection and Preparation of Influent, Effluent, and Sewage Canal Samples

Influent samples were collected at the drain that receives both water and livestock waste from pig pen cleaning activities. Effluent samples were collected at the effluent tank, which was 20 cm in depth and situated in the center of biogas plant [21]. The effluent tank of the biogas plant is where the farmers collect effluent for irrigation of agriculture/horticulture. Sewage canal samples were collected at the open household drain that receives biogas effluent, wastewater, and other runoff flow. For sampling and concentration, composite samples consisting of three individual 50 mL samples were placed in a 50 mL sterile falcon tube [19]. All of the samples were stored in a cooling box and transported to the laboratory at the National Institute of Hygiene and Epidemiology (NIHE) in Hanoi on the day of sampling for further processing. Samples were analyzed according to a previously described method. Briefly, 10 g of composite waste samples was added to 90 mL of distilled water and homogenized for 30 s in a vortex mixer (Pulsifier^®^, Filtaflex, Almonte, ON, Canada). Samples were concentrated by centrifugation followed by a flotation step where 10 mL of the pelleted sample was underlaid with 5 mL of flotation fluid (saturated NaCl solution with 500 g of glucose added per liter; diluted 1:1 with sterile distilled water to a final specific gravity of 1.13 g/mL), and centrifuged for 1 min at 100× *g*. All of the supernatant was transferred to a clean tube to remove larger debris. The sample was subsequently washed twice with sterile distilled water with centrifugation of 1540× *g* for 10 min to remove the remains of the flotation fluid before concentrating the sample volume to 2 mL [19,28].

#### 2.4.2. Collection and Preparation of Surface Water Samples

Surface water samples (lake, pond or river), which were in close proximity to a biogas plant or received biogas wastewater runoff that was used for irrigation purposes, were collected (Figure 3). Composite water samples consisting of three individual 1 L samples were collected in sterile, wide-mouth, screw-capped 1 L bottles. All of the samples were stored in a cooling box and transported to the NIHE laboratory on the day of sampling. Further analysis was implemented by concentration and centrifugation followed by a flotation step with saturated NaCl–glucose solution to attain the final volume of 2 mL, as described above [19,28].

#### 2.4.3. Collection and Preparation of Vegetables and Soils

Vegetables grown in household gardens and soils, which were close to a biogas plant or received biogas waste, were randomly collected. Two hundred grams of vegetables (stem and leaves) or soils were picked using sterile plastic bags. Then, the samples were stored in a cooling box and transported to NIHE on the day of sampling. Samples of 10 g of vegetables (stems and leaves) or 10 g of soils were washed with 90 mL of 0.01% TWEEN20 (Sigma-Aldrich, Merck, Darmstadt, Germany) in distilled water in a Pulsifier instrument (Pulsifier^®^, Filtaflex, Almonte, ON, Canada). Samples were concentrated by centrifugation followed by a flotation step, where 10 mL of the pelleted sample was underlaid with 5 mL of flotation fluid (saturated NaCl solution with 500 g of glucose added per liter; diluted 1:1 with sterile distilled water to a final specific gravity of 1.13), and centrifuged for 1 min at 100× *g.* All of the supernatant was transferred to a clean tube to remove larger debris The sample was subsequently washed twice with sterile distilled water to remove the remains of the flotation fluid before concentrating the sample volume to 2 mL [19,28].

### 2.5. Enumeration of Cryptosporidium and Giardia (oo)cysts

Ten microliters of the concentrated sample was microscopically examined. Oocysts and cysts were detected and identified in a 10 µL solution (20 µL Merifluor^®^
*Cryptosporidium*/*Giardia* kit (Meridian Bioscience, Inc., Newtown, OH, USA), 0.5 µL DAPI (4′,6-diaminodino-2-phenyl-indol, Sigma-Aldrich, St. Louis, MO, USA), 2 mg/mL and 80 µL PBS 1×) mixed with a 10 µL mounting buffer. *Giardia* and *Cryptosporidium* (oo)cysts were quantified using epifluorescent microscopy (Nikon, Eclipse E600) with a magnification of 300–800×. A blue filter (excitation, 480 nm; emission, 520 nm) was used to detect FITC-conjugated MAb-labelled (oo)cysts and a UV filter block was used for DAPI (excitation, 350 nm; emission, 450 nm). The microorganism identification was based on their morphology, fluorescence, and size [29]. The number of (oo)cysts was calculated/100 mL for influent samples, effluent samples, sewage canal samples, and surface water samples and 100 g for soil and vegetable samples.

The collected data were used to quantify the level of contamination of *Cryptosporidium* and *Giardia* (oo)cysts in environmental samples. Based on this information, the concentration of oo(cysts) was calculated according to an equation referenced from Adeyemo et al. in 2019 [30].
$$\mathrm{Concentration}((\mathrm{oo})\mathrm{cysts}\;\mathrm{per}\;100\;\mathrm{mL})=\frac{N}{V}$$
where N is the number of (oo)cysts detected and V is the volume of processed sample (100 mL or 100 g).

### 2.6. Statistical Analysis

Statistical analysis was performed using STATA/MP software version 16.0 (StataCorp LLC, College Station, TX, USA). Both descriptive and inferential statistics were used. Descriptive statistics were performed on qualitative variables (frequency and percentage value calculations) and quantitative variables (median, min, and max value calculations). Inferential statistics of the Pearson Chi-Square test and Fisher’s Exact test were applied where appropriate to compare the differences of social characteristics of the study population and the practices of biogas wastewater management by farmers in Bac Giang. Differences were set as statistically significant at a probability level of *p* < 0.05.

## 3. Results and Discussion

The analysis of food and environmental samples for parasites of public health significance, including *Cryptosporidium* and *Giardia*, has been the subject of several previous studies [2,10,31] and our survey contributes further data. In the present study, a total of 239 samples were collected from different types of samples in the studied districts in Bac Giang province in 2020 (Figure 1 and Figure 4; Table 1) via direct immunofluorescent detection to study the occurrence of *Cryptosporidium* spp. and *Giardia* spp. (oo)cysts. Among the samples, 19 were *Cryptosporidium* positive, and 40 were *Giardia* positive. The percentages of positive detection were 7.9% (95% CI: 4.5–11.4) (19/239) and 16.7% (95% CI: 12.0–21.5) (40/239), respectively. In detail, the results indicated that the percentages of positive detection of *Cryptosporidium* spp. and *Giardia* spp. in influent, effluent, sewage canal, and vegetables were 13.1% (11/84), 6.0% (5/83), 15.4% (2/13) and 5.9% (1/17) and 26.2% (22/84), 7.2% (6/83), 7.7% (1/13) and 5.9% (1/17), respectively. *Cryptosporidium* and *Giardia* (oo)cysts were not detected in lake/pond water and soil samples. The results indicate that these environmental materials are contaminated with *Cryptosporidium* and *Giardia* (oo)cysts, and thus can act as a contamination source for human and animal infections in the area. The variation in occurrence of (oo)cysts was examined in different sampling locations before and after biogas plants. The results confirmed that *Cryptosporidium* spp. and *Giardia* spp. (oo)cysts are the most common in 11 (13.1%) vs. 22 (26.2%) influent samples followed by 5 (6.0%) vs. 16 (19.3%) effluents, as presented in Figure 4. *Cryptosporidium* spp. were more commonly found in sewage canals, while *Giardia* spp. were more prevalent in biogas influent and effluent samples. However, the prevalence of *Giardia* spp. decreased gradually depending on the proximity of biogas plant. Significantly more *Giardia* cysts were detected than *Cryptosporidium* oocysts in our survey of biogas influent and effluent samples, which has been also indicated by clinical observations where giardiasis is more frequently reported than cryptosporidiosis [32,33]. In 2014, the same research approach was carried out by Kitajima et al. [34] in Arizona (USA), whose aim was to detect *Giardia* and *Cryptosporidium* in the influent and effluent of two WWTPs (activated sludge systems). In this case, *Giardia* was more abundant than *Cryptosporidium* in about 1–2 log for both influent and effluent. The greater detection of *Giardia* cysts may be due to the infection intensity, zoonotic infection or other host-parasite, environmental factors, and hydrophobic tendency [35].

The concentrations of *Giardia* spp. cysts and *Cryptosporidium* spp. oocysts in the samples of biogas influents, effluents, sewage canals receiving biogas wastewater, and vegetables in all three districts are presented in Table 1. Of note, *Giardia* spp. cyst concentration varied from 1 × 10^4^ to 2 × 10^6^ cysts/100 mL of biogas influent and from ‘not detected’ to 5.5 × 10^4^ cysts/100 mL of biogas effluent. *Cryptosporidium* spp. oocyst concentration varied from 1 × 10^4^ to 3 × 10^5^ oocysts/100 mL of biogas influent and from ‘not detected’ to 2 × 10^5^ oocysts/100 mL of biogas effluent. Both *Giardia* spp. cysts and *Cryptosporidium* spp. oocysts were found in lowest concentrations in sewage canals, varying from ‘not detected’ to 2 × 10^4^ (oo)cysts/100 mL. Although the concentration of (oo)cysts was lower in effluents than in raw influents, effluents from biogas plants are still more commonly used in agriculture than raw influents and thus, may pose a potential health risk to animals and humans, especially in more sensitive populations, such as children, the elderly, and immunocompromised. *Cryptosporidium* and *Giardia* (oo)cysts are very resilient and may survive in water for months and in particular *Cryptosporidium* may survive for a long time. In 2014, Huong et al. [36] identified that more than 90% of pig farms in Vietnam used disinfectants, such as chloramine, iodine, and lime to clean and disinfect pig pens. It is unknown to what extent the use of disinfectants can affect the microbial populations. However, *Cryptosporidium* and *Giardia* (oo)cysts are well known to be very resistant to chemical disinfection [30]. As a result, even at low concentrations, these parasites are worrisome, due to the high observed risk of infection, as shown by Sato et al. in 2013 [37]. Several studies indicated that lake/pond water was found to harbor *Cryptosporidium* and *Giardia* (oo)cysts [38,39,40]. This is contrary to our presented results where we found that the frequency and counts of both *Giardia* and *Cryptosporidium* (oo)cysts were zero in the investigated samples of lakes/ponds and soils. This may be due to the low recovery of the (oo)cysts, which was potentially below the detection limit. The inability to detect frequency or counts below the detection limit is a recurrent issue when dealing with environmental samples and is caused by the low concentration of an agent in the environment or low recovery of the applied methodology due to the various factors influencing the samples and methodology quality [30]. Although the obtained concentrations of these parasites were low or under the detection limit, their prevalence is high in the global population [9,41], as these protozoan parasites are responsible for several outbreaks worldwide [5,6,7]. In a study on the use of tubular digesters to treat livestock waste, it was found that *C. parvum* posed a greater risk than *G. lamblia* in all of the exposure pathways (fomite, soil, and crop contamination from runoff) due to livestock shedding higher loads of *Cryptosporidium* oocysts and lower inactivation rates of *Cryptosporidium* oocysts during anaerobic digestion in comparison to *G. lamblia* cysts [41]. The risk of infection from exposure to contaminated soil and crops was significantly lower for a community using tubular digesters to treat livestock waste compared to a community where the untreated waste was applied to soil [42]. With the increased awareness of protozoan disease outbreaks, more attention should be paid in the future to the mitigation of risks associated with protozoa that emerged from biogas waste.

Vegetables are important in the consumer’s dietary habit in almost all countries. It has been indicated that protozoan contamination of foods of plant origin may constitute a potential health hazard if these kinds of foods, which are normally consumed with minimal process, are ready to eat vegetables or raw and salads prepared in poor hygiene practices [43,44]. Regarding the limitation of water resort, the reuse of treated/untreated wastewater to irrigate farmlands has been highlighted [45]. However, the use of untreated wastewater or biogas effluents as a fertilizer for different crops is the main source of contamination as the demand for vegetables and fruits increases [46,47,48]. In our study, both *Cryptosporidium* and *Giardia* (oo)cysts were found in lowest concentration varying from ‘not detected’ to 2.10^4^ (oo)cysts/100 g with prevalence of 7.1% (1/17) vs. 5.9% (1/17) in vegetables. In a systematic review and meta-analysis conducted by Karshima in 2018 [49], *Cryptosporidium* was thought to be the most prevalent parasite from fruits and vegetables in Nigeria. Utaaker et al. in 2017a [50] identified that *Cryptosporidium* and *Giardia* (oo)cysts are among 6% (17/284) and 5% (13/284) of vegetable samples in India. Alemu et al. in 2019 [51] reported the presence of *Cryptosporidium* and *Giardia* in 5.8% and 6.9% of 347 vegetable samples collected from markets in southern Ethiopia. In a study performed by Eraky et al. in 2014 [52] in Egypt, from 530 vegetable samples, *Giardia* with the prevalence rate of 8.8% was the most prevalent parasite detected from the samples. In China, the study of 642 market vegetables indicated that *Cryptosporidium* spp. was responsible for 16 (2.5%), while *Giardia* spp. accounted for 73 (14%) [47,53]. Moreover, Utaaker et al. in 2017b [54] reported that *Cryptosporidium* and *Giardia* could remain viable on lettuce leaves and increased the chance of contamination in consumers. Therefore, in addition to increasing the irrigation of farms using raw wastewater, the risk of parasitic contamination can be increased.

In this study, few numbers were of farmers connected their toilet to the biogas unit. This was mainly due to the fact that the flush toilet had already been installed before the installation of biogas units. However, open sewage canals next to biogas units were receiving both biogas wastewater and flush toilet wastewater. These wastewater sources were used as a fertilizer for different crops. More farmers in Lang Giang (19.3%) used this wastewater for vegetables eaten raw in comparison to farmers in Hiep Hoa (9.2%), respectively. It appears from the interviews that biogas wastewater was mainly used to fertilize fruit trees with 37% in Lang Giang, 25.6% in Viet Yen, and 20.9% in Hiep Hoa. In 2010, Rzeżutka et al. [55] analyzed the presence of oocysts of *Cryptosporidium* in vegetables grown in areas with moderate to high livestock production and claimed that close contact between vegetable farms and animal husbandry may enhance the risk of contamination of vegetables. This is of concern in Lang Giang, Hiep Hoa, and Viet Yen districts, where the main source of irrigation water in these areas is surface water bodies or biogas effluents, which can bring *Cryptosporidium* and *Giardia* (oo)cysts from faeces of humans and livestock to downstream vegetable farmlands. *Cryptosporidium* and *Giardia* (oo)cysts are buried with animal faeces and can be distributed to vegetable farmlands. In addition, they can remain infective for a long time due to their resistant wall [50]. Furthermore, due to the large livestock development in these regions, manure as human and animal product, played a critical role in fertilizing the farmlands.

There are studies indicating the insufficiency of the routine treatment processes used in WWTPs for the removal of all (oo)cysts of *Cryptosporidium* and *Giardia* [9,56,57,58]. In 2013, a study published by Sroka et al. [59] showed that the numbers of viable (oo)cysts of parasites are not reduced during the treatment process. In this study, a research for the main treatment method for raw livestock manure in rural communities, was carried out at biogas plants. Most of the biogas plants in Bac Giang were installed in a period of 5–10 years. They have an average volume of 13 cubic meters. All of the households, predominantly those with small-medium scale pig farms (10–50 pigs) used pig manure in their biogas plant. Poultry was also raised on a medium scale with less than 100 chickens per farm and very few households discharged poultry manure to the biogas plants (Table 2). Nearly all of the farmers acknowledged that the installation of the biogas system had direct benefits on the daily lives of poor farming households. Most importantly, it allowed for cleaner kitchens and reduced indoor air pollution. However, strong bad odors associated with pig manure were particularly seen as a nuisance to the household members and neighbors. As observed in this study, strong bad odors were due to the insufficient treatment of biogas plants. More specifically, insufficient treatment refers to the following: (1) No maintenance was carried out during their usage; (2) infrequent cleaning of biogas plants, with 30.4% of farmers in Lang Giang, 27.9% of farmers in Viet Yen, and 45.9% of farmers in Hiep Hoa; and (3) over usage of biogas plant capacity. Using personal protective measures when handling biogas wastewater was prevalent (40–65%) in the interviewed participants (Table 3). Amongst the interviewed individuals, 30–60% reported wearing a face mask, 40–70% reported wearing a boost, and 40–60% wore gloves and washed their hands with soap. Only 10% of the participants always used all four of the aforementioned personal protective measures when handling biogas wastewater while cleaning biogas plants or performing agricultural activities. It is unfortunate that our study did not measure the association between health risk and the use of protective measures while participating in activities exposed to biogas plants. However, in 2006, Trang do et al. [18] indicated that poor sanitation and hygiene practices and the lack of use of protective measures are important independent risk factors for helminth infections in Vietnam. In addition, a higher risk of diarrhea was found among people with inadequate use of protective measures in a study on the practices of farmers in agriculture activities in Hanam [60]. Moreover, the cleaning of a biogas plant is considered an exposure event, especially for farmers, who may not be wearing appropriate protective equipment. There is a need to raise awareness on protective measures in order to minimize potentially negative consequences for public health.

## 4. Conclusions

In the present study, the results revealed the occurrence of *Cryptosporidium* and *Giardia* in biogas waste and their presence in vegetables in Bac Giang, Vietnam. In addition, the study emphasized the importance of effective management of livestock manure by biogas plants in Vietnam.

The limitation of this study was the lack of molecular characterization of the identified (oo)cysts since the public health importance and clinical significance of the parasites depend on the specific species, genotypes, and their viability or infectivity to cause human infection.

Further actions to improve the quality of biogas effluents are required as the pathogenic species of *Cryptosporidium* and *Giardia* (oo)cysts can result in potential risk for zoonotic and anthroponotic transmission. This is due to the exposure to biogas waste or contamination of food of plant origin, in the case that these kinds of food are normally consumed raw or with minimal processing.

Moreover, there is a need for the education of local governments and health authorities by offering them the possibility to understand the health risks associated with the exposure to biogas wastewater and to undertake control measures.

Furthermore, it is imperative that these technologies have the ability to reduce the contamination of water and food sources by these two pathogenic protozoa and thus, protect the health of animals and humans in the context of One Health concept.

## Figures and Tables

**Figure 1 pathogens-11-00174-f001:**
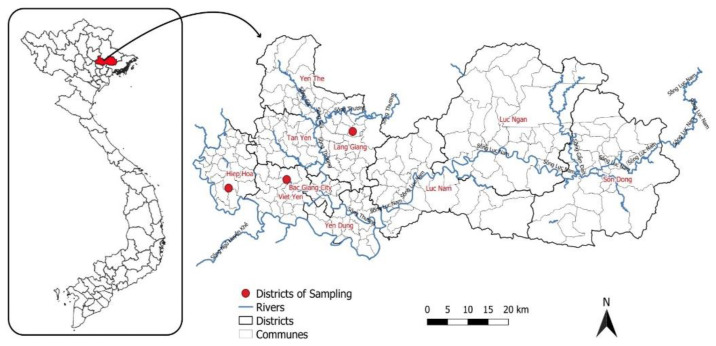
Map of Bac Giang province, where the red dots are illustrated for districts of sampling.

**Figure 2 pathogens-11-00174-f002:**
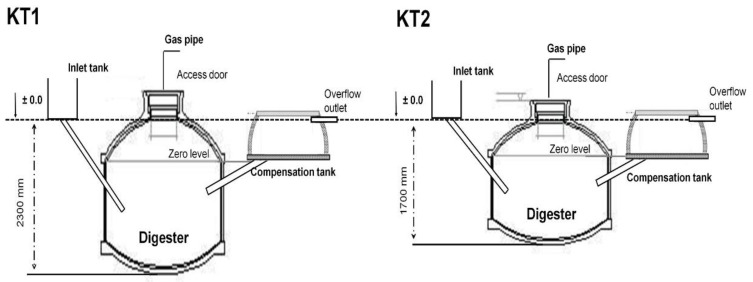
Schematic drawing of biogas plant types (KT1 and KT2) used by farmers in Bac Giang province).

**Figure 3 pathogens-11-00174-f003:**
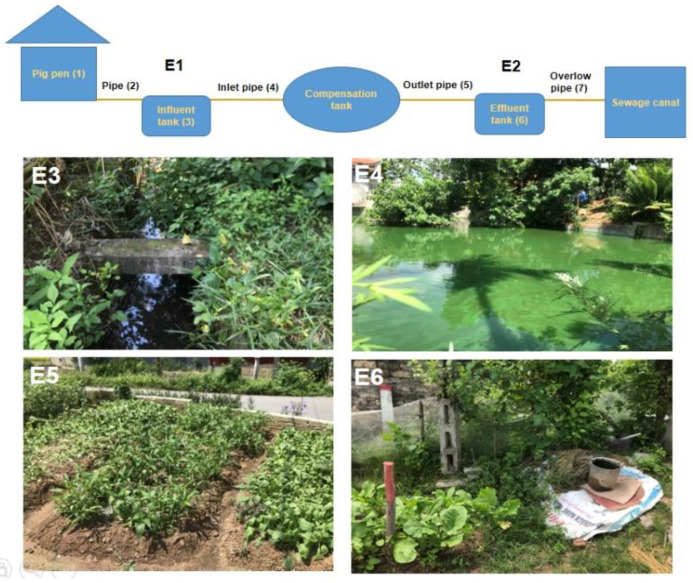
Schematic drawing of sampling locations (*Source: Vu Van Tu* and *Nguyen Manh Hien*). E1-Influent (composite sample of 2, 3, 4); E2-Effluent (composite sample of 5, 6, 7); E3-Sewage canal next to biogas plants and receiving both biogas and flush toilet wastewater; E4-Lake/pond in a close area of biogas plants; E5-Soil near biogas plants and receiving biogas waste; E6-Vegetables grown near biogas plants and receiving biogas waste for irrigation.

**Figure 4 pathogens-11-00174-f004:**
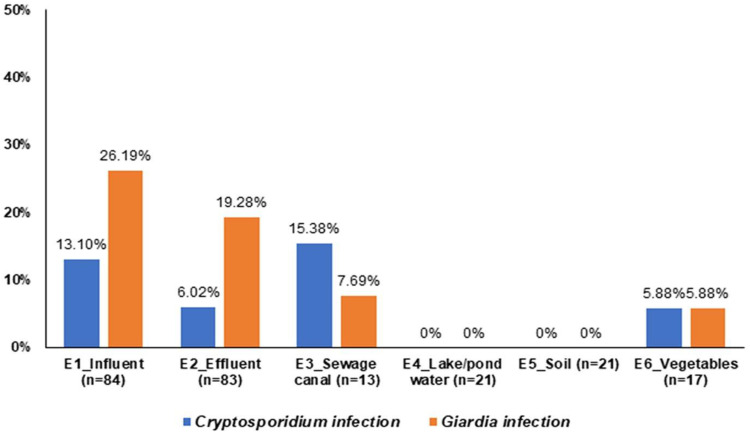
Variation of prevalence of *Cryptosporidium* and *Giardia* (oo)cysts in different sampling locations derived from the biogas system in Bac Giang province.

**Table 1 pathogens-11-00174-t001:** Occurrence and concentration of *Giardia* spp. cysts and *Cryptosporidium* spp. oocysts/100 mL of biogas influents, effluents, sewage canal samples, and 100 g of vegetables by district.

Area/ (n = Number of Samples)	Lang Giang (N = 135)		Viet Yen (N = 172)		Hiep Hoa (N = 87)	
Pathogen Type of Sample	*Giardia* spp.	*Cryptosporidium* spp.	*Giardia* spp.	*Cryptosporidium* spp.	*Giardia* spp.	*Cryptosporidium* spp.
Influent (E1, n = 84)						
Positive sample/examined (%)	2/11 (18.8)	1/11 (9.1)	14/57 (24.6)	9/57 (15.8)	6/16 (37.5)	1/16 (6.3)
Median number of parasites	2.55 × 10^5^	3 × 10^5^	1 × 10^4^	2 × 10^4^	1.75 × 10^5^	2 × 10^5^
Min–Max ^a^	1 × 10^4^–5 × 10^5^	3 × 10^5^–3 × 10^5^	1 × 10^4^–2 × 10^6^	1 × 10^4^–1.4 × 10^5^	1 × 10^4^–4.1 × 10^4^	2 × 10^5^–2 × 10^5^
Effluent (E2, n = 83)						
Positive sample/examined (%)	ND	1/11 (9.1)	11/56 (19.6)	ND	5/16 (31.3)	4/16 (25.0)
Median number of parasites		2 × 10^5^	2 × 10^5^		1 × 10^5^	1.5 × 10^4^
Min–Max ^a^		2 × 10^5^–2 × 10^5^	1 × 10^4^–8 × 10^4^		4 × 10^4^–5.5 × 10^5^	1 × 10^4^–2 × 10^4^
Sewage canal (E3, n = 13)						
Positive sample/examined (%)	ND	ND	1/11 (9.1)	2/11 (18.2)	ND	ND
Median number of parasites			1 × 10^4^	1 × 10^4^		
Min–Max ^a^			1 × 10^4^–1 × 10^4^	1 × 10^4^–1 × 10^4^		
Vegetables (E6, n = 17)						
Positive sample/examined (%)	ND	ND	1/17 (5.9)	1/17 (7.1)	ND	ND
Median number of parasites			1 × 10^4^	2 × 10^4^		
Min–Max ^a^			1 × 10^4^–1 × 10^4^	2 × 10^4^–2 × 10^4^		

^a^ The median values were calculated based on the number of positive samples. Note: ND—not detected.

**Table 2 pathogens-11-00174-t002:** Characteristics of the study population in Bac Giang province, Vietnam.

Characteristics	Lạng Giang	Viet Yen	Hiep Hoa	
	N = 135	N = 172	N = 87	*p*-Value
Area used for agriculture (1000–3000 m^2^)	112 (84.2%)	96 (61.9%)	47 (55.3%)	<0.001 ^a^
Age group (41–54 years old)	65 (48.1%)	77 (44.8%)	45 (51.7%)	0.56 ^a^
Males	107 (79.3%)	133 (77.3%)	66 (75.9%)	0.83 ^a^
High education level	1 (0.7%)	2 (1.2%)	2 (2.3%)	0.22 ^b^
Farmer	134 (99.3%)	169 (98.3%)	84 (96.6%)	0.18 ^b^
Size of the household (≥5 people)	72 (53.3%)	91 (52.9%)	49 (56.3%)	0.87 ^a^
No. of pigs raised				<0.001 ^a^
<10 pigs	65 (48.2%)	35 (20.4%)	18 (20.7%)	
10–50 pigs	58 (43.0%)	121 (70.4%)	59 (67.8%)	
>50 pigs	12 (8.9%)	16 (9.3%)	10 (11.5%)	
No. of poultry raised				<0.001 ^a^
No poultry	5 (3.7%)	17 (9.9%)	6 (6.9%)	
<100	117 (86.7%)	140 (81.4%)	59 (67.8%)	
100–200	9 (6.7%)	10 (5.8%)	8 (9.2%)	
>200	4 (2.9%)	5 (2.9%)	14 (16.1%)	

^a^*p*-value based on the Pearson Chi-Square test. ^b^*p*-value based on Fisher’s Exact test.

**Table 3 pathogens-11-00174-t003:** Practices of biogas wastewater management by farmers in Bac Giang, Vietnam.

	Lang Giang	Viet Yen	Hiep Hoa	
	N = 135	N = 172	N = 87	*p*-Value
5–10 years of age of biogas plant	72 (53.3%)	106 (61.6%)	48 (55.2%)	0.002 ^a^
Clean-up of biogas plant	41 (30.4%)	48 (27.9%)	40 (45.9%)	0.011 ^a^
No repair of biogas plant	126 (93.3%)	168 (98.2%)	85 (97.7%)	0.08 ^b^
Pig manure used for biogas plant	134 (99.3%)	170 (98.8%)	87 (100%)	0.80 ^b^
Poultry manure used for biogas plant	1 (0.7%)	3 (1.7%)	0 (0%)	0.55 ^b^
Possible opened effluent tank	123 (91.1%)	142 (83.0%)	83 (95.4%)	0.002 ^b^
Household involved in effluent tank cleaning	29 (21.5%)	36 (20.9%)	20 (22.9%)	0.25 ^b^
Household involved in sewage canal cleaning	72 (53.3%)	80 (46.5%)	37 (42.5%)	0.26 ^a^
Personal protective measures				
Wearing face mask	48 (62.3%)	30 (31.3%)	21 (45.7%)	<0.001 ^b^
Wearing boost	50 (64.9%)	35 (37.2%)	22 (47.8%)	<0.001 ^b^
Wearing gloves	42 (54.6%)	29 (31.5%)	18 (39.1%)	0.001 ^a^
Washing hands with soap	43 (56.6%)	33 (34.4%)	18 (39.1%)	<0.001 ^a^
Use of biogas wastewater				
Fertilizer of corns	27 (20.0%)	22 (12.8%)	6 (6.9%)	0.052 ^b^
Fertilizer of fruit tree	50 (37.0%)	44 (25.6%)	18 (20.9%)	0.025 ^b^
Fertilizer of vegetables	26 (19.3%)	23 (13.5%)	8 (9.2%)	0.12 ^b^

^a^*p*-value based on the Pearson Chi-Square test. ^b^*p*-value based on Fisher’s Exact test.

## Data Availability

Not applicable.

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
