# Peer review of "Cryptosporidium and Giardia in Biogas Wastewater: Management of Manure Livestock and Hygiene Aspects Using Influent, Effluent, Sewage Canal Samples, Vegetable, and Soil Samples"

_pathogens, 2022, doi:10.3390/pathogens11020174_

Round 1

Reviewer 1 Report

The manuscript describes results from the study of the Cryptosporidium spp. and Giardia spp. (oo)cysts occurrence in biogas wastewater.

The article is very interesting for the parasitologist as well as the epidemiologist. However, I have few minor issues which should be improved/considered by the authors:

  1. Line 43 citation is needed
  2. Line 63-87: in my opinion these part fits better to the discussion than intoduction. The autors should shorten it or move it into discussion.
  3. Line 116-122: The authors should consider to deleting these part because it sounds like conclusions.

Author Response

Point by point response to reviewers’ comments

Reviewer #1

General comments:

The manuscript describes results from the study of the Cryptosporidium spp. and Giardia spp. (oo)cysts occurrence in biogas wastewater.

Specific comments:

The article is very interesting for the parasitologist as well as the epidemiologist. However, I have few minor issues which should be improved/considered by the authors:

Line 43 citation is needed

Response

We appreciate the comment. The citation is added [line 49]

Line 63-87: in my opinion these part fits better to the discussion than introduction. The authors should shorten it or move it into discussion.

Response

Thanks for your comment. The paragraph has been shorten as suggested [line 63-80].

Line 116-122: The authors should consider to deleting these parts because it sounds like conclusions.

Response

We agree with your comment. These parts have been deleted and now changed as indicated in the text [line 116 – 119].

Reviewer #2

General comment

The manuscript by Tram et al. is undoubtedly original, well written and conducted, with a relevant contribution to the understanding of the complex epidemiology puzzle of Cryptosporidium and Giardia circulation in the environment. I support its possible publication after appropriate modifications as outlined below.

Specific comments

The references citations within the whole manuscript are not in agreement with the journal requirement. Also, there are some discrepancies concerning the font size of some paragraph. Please be carefully with this concern!

Response

We appreciate the comment. The references citations and font size discrepancies have been edited and revised within the whole manuscript as suggested.

Lines 39-49: please insert appropriate references to support the statements

Response

We appreciate the comment. The citation is added [line 49]

Line 54: oo(cysts) - being the first appearance in the text please highlight for the reader that Cryptosporidium is transmitted through oocysts and Giardia through cyst

Response

Thank for your comment. We have changed the text with highlight for the reader that Cryptosporidium  is transmitted through oocysts and Giardia through cysts [line 53-54]

Lines 58-59: unclear sentence, please rephrase it

Response

Thank for your comment. We have rephrased the sentence as suggested [line 58-59]

Lines 63-73: the authors need to compress these lines. In the present form seems to be “Discussion” rather than “Introduction”. The presented detailed information can be incorporated in the discussion chapter. The same requirement for lines 80-88

Response

The paragraph has been shortened as recommended [line 51-64]

Line 73: new paragraph “The occurrence of ..”

Response

We appreciate the comment. The new paragraph “The occurrence of... “ has been made as suggested [line 76-82]

Line 77: insert the appropriate reference after the sentence

Response

Thank for your comment. The appropriate reference has been inserted [line 77]

Line 91: please insert a reference after the sentence

Response

Thank for your comment. The appropriate reference has been inserted [line 93]

Line 96: to be more explicit - …Hanoi and Hanam, Vietnam.

Response

Thank for your comment. The sentence has been changed to be more explicit “.... Hanoi and Hanam. Vietnam” [line 99]

Lines 113-122: I suggest to rewrite this paragraph, presenting only the aim of the study. The other mentioned issues can be included in the Introduction chapter before this paragraph.

Response

We appreciate the comment. We have re-written the paragraph as suggested [line 119-122].

Lines 163-174: to increase the reader interest I suggests the inclusion of the used structural questionnary as supplementary file.

Response

Thank for the comment.

Lines 176-177: Please justify your choice of number of collected samples in the study. In this regard, the authors need to refer to a statistical model, based on which they can validate the survey results. So, the authors must convince the scientific community that they results are completely supportable by statistical tools.

Response

Thank for your comment. We have added the appropriate reference [line 174].

Line 176: I wonder about the reason of the resulted slight discrepancy between the “84 influent samples, 83 effluent samples”? I suggest the presentation of sample collection in a single subchapter (union of 2.4 and 2.5)

Response

We appreciate the comment. There was a missing one effluent sample due to the inability of accessing the effluent tank during sampling at household. We have kept the section 2.4 as general presentation of sample collection and subsection 2.4.1 – 2.4.3 describing the detailed collection of different type of samples.

Lines 193-195: “For sampling and concentration, a composite sample consisting of three individual 50 mL samples were in a 50 mL sterile falcon tubes.” – please indicate a reference using this type of sample collection

Response

Yes, we agree. The appropriate reference has been added [line 197, 201]

Line 206: please insert a reference after this paragraph

Response

Thank for your comment. The appropriate reference has been added for the paragraph [line 212-213]

Line 211: please insert a reference indicating this type of sample collection (surface water samples)

Response

Thank for your comment. The appropriate reference has been added for the paragraph (surface water sample) [line 221]

Line 218: please insert a reference indicating this type of sample collection (vegetable and soil samples)

Response

Thank for your comment. The appropriate reference has been added for the paragraph (vegetable and soil sample) [line 233-234]

Line 258: in order to increase the reader interest, I suggest the inclusion of representative pictures about positive immunofluorescence findings as supplementary material

Response

We appreciate the comment.

Lines 260-261: when you express overall prevalence value results, please indicate the 95% Confidence Interval in brackets

Response

We appreciate the comment. We have added the 95% CI in brackets [line 260-261]

Lines 298-301, 387-389, 397-398, first column of the Tables 2, 3: formatting error – please revise these sentences. Please be carefully with these concerns throughout the manuscript

In the paragraph starting with the line 310: - in order to improve the discussion chapter and contrary to the presented results, referring to the lack of the detection of Cryptosporidium and Giardia (oo)cyst in lake/pond water the authors need to refer to other studies presenting positive findings (e.g. PMID: 19705155; PMID: 28832257; PMID: 16872371), and highlight that these surface water types can also harbor (oo)cysts. These articles can be consulted and cited (!).

Response

We appreciate the comment. Formatting errors have been revised as indicated in the text [line 297-404] and first column of the table 2 and 3.

Referring to the lack of the detection of Cryptosporidium and Giardia (oo)cysts in lake/pond water, we highlight surface water types can also harbour (oo)cysts and the recommended references have been added accordingly [line 315-319]

Lines 408-409: please compress in a single sentence

Response

Thank for the comment. We have compressed in a single sentence [line 405-407]

Also, in the conclusion section the authors need to highlight as study limitation, the lack of the molecular characterization of the identified (oo)cyst, to evaluate the really threat for the public health.

Response

Yes, we agree, and we have highlighted the lack of molecular characterization of the identified (oo)cysts as study limitation [line 419-420].

Reviewer 2 Report

The manuscript by Tram et al. is undoubtedly original, well written and conducted, with a relevant contribution to the understanding of the complex epidemiology puzzle of Cryptosporidium and Giardia circulation in the environment. I support its possible publication after appropriate modifications as outlined below:

General comment

The references citations within the whole manuscript are not in agreement with the journal requirement. Also, there are some discrepancies concerning the font size of some paragraph. Please be carefully with this concern!

Specific comments

Lines 39-49: please insert appropriate references to support the statements

Line 54: oo(cysts) - being the first appearance in the text please highlight for the reader that Cryptosporidium is transmitted through oocysts and Giardia through cyst

Lines 58-59: unclear sentence, please rephrase it

Lines 63-73: the authors need to compress these lines. In the present form seems to be “Discussion” rather than “Introduction”. The presented detailed information can be incorporated in the discussion chapter. The same requirement for lines 80-88

Line 73: new paragraph “The occurrence of ..”

Line 77: insert the appropriate reference after the sentence

Line 91: please insert a reference after the sentence

Line 96: to be more explicit - …Hanoi and Hanam, Vietnam.

Lines 113-122: I suggest to rewrite this paragraph, presenting only the aim of the study. The other mentioned issues can be included in the Introduction chapter before this paragraph.

Lines 163-174: to increase the reader interest I suggests the inclusion of the used structural questionnary as supplementary file.

Lines 176-177: Please justify your choice of number of collected samples in the study. In this regard, the authors need to refer to a statistical model, based on which they can validate the survey results. So, the authors must convince the scientific community that they results are completely supportable by statistical tools.

Line 176: I wonder about the reason of the resulted slight discrepancy between the “84 influent samples, 83 effluent samples”?

I suggest the presentation of sample collection in a single subchapter (union of 2.4 and 2.5)

Lines 193-195: “For sampling and concentration, a composite sample consisting of three individual 50 mL samples were in a 50 mL sterile falcon tubes.” – please indicate a reference using this type of sample collection

Line 206: please insert a reference after this paragraph

Line 211: please insert a reference indicating this type of sample collection (surface water samples)

Line 218: please insert a reference indicating this type of sample collection (vegetable and soil samples)

Line 258: in order to increase the reader interest, I suggest the inclusion of representative pictures about positive immunofluorescence findings as supplementary material

Lines 260-261: when you express overall prevalence value results, please indicate the 95%Confidence Interval in brackets

Lines 298-301, 387-389, 397-398, first column of the Tables 2, 3: formatting error – please revise these sentences. Please be carefully with these concerns throughout the manuscript

In the paragraph starting with the line 310: - in order to improve the discussion chapter and contrary to the presented results, referring to the lack of the detection of Cryptosporidium and Giardia (oo)cyst in lake/pond water the authors need to refer to other studies presenting positive findings (e.g. PMID: 19705155; PMID: 28832257; PMID: 16872371), and highlight that these surface water types can also harbor (oo)cysts. These articles can be consulted and cited (!).

Lines 408-409: please compress in a single sentence

Also, in the conclusion section the authors need to highlight as study limitation, the lack of the molecular characterization of the identified (oo)cyst, to evaluate the really threat for the public health.

Author Response

Thank you for your suggestion

Round 2

Reviewer 2 Report

The authors correctly acknowledged almost all of the raised concerns.

Only two minor changes are further necessary namely:

  • please completely de-italicize the first column content of tables 2 and 3 in order to distinguish the scientific name of species within the manuscript;
  • please clearly indicate on figures of the supplementary material the identity of the Cryptosporidium oocysts and Giardia cysts, and also insert a scale bar;
